# Greenland subglacial drainage evolution regulated by weakly connected regions of the bed

Matthew J. Hoffman[1], Lauren C. Andrews[2], Stephen A. Price[1], Ginny A. Catania[3,4], Thomas A. Neumann[2], Martin P. Lüthi[5], Jason Gulley[6], Claudia Ryser[7], Robert L. Hawley[8] & Blaine Morriss[9]

Penetration of surface meltwater to the bed of the Greenland Ice Sheet each summer causes an initial increase in ice speed due to elevated basal water pressure, followed by slowdown in late summer that continues into fall and winter. While this seasonal pattern is commonly explained by an evolution of the subglacial drainage system from an inefficient distributed to efficient channelized configuration, mounting evidence indicates that subglacial channels are unable to explain important aspects of hydrodynamic coupling in late summer and fall. Here we use numerical models of subglacial drainage and ice flow to show that limited, gradual leakage of water and lowering of water pressure in weakly connected regions of the bed can explain the dominant features in late and post melt season ice dynamics. These results suggest that a third weakly connected drainage component should be included in the conceptual model of subglacial hydrology.

[1] Fluid Dynamics and Solid Mechanics Group, Los Alamos National Laboratory, Los Alamos, New Mexico 87545, USA. [2] Cryospheric Sciences Laboratory, NASA Goddard Space Flight Center, Greenbelt, Maryland 20771, USA. [3] Institute for Geophysics, Jackson School of Geosciences, The University of Texas at Austin, Austin, Texas 78758, USA. [4] Department of Geological Sciences, Jackson School of Geosciences, The University of Texas at Austin, Austin, Texas 78758, USA. [5] Glaciology and Geomorphodynamics Group, Department of Geography, University of Zürich, 8057 Zürich, Switzerland. [6] School of Geosciences, University of South Florida, Tampa, Florida 33620, USA. [7] Laboratory of Hydraulics, Hydrology and Glaciology, Swiss Federal Institute of Technology (ETH) Zürich, 8093 Zürich, Switzerland. [8] Department of Earth Sciences, Dartmouth College, Hanover, New Hampshire 03755, USA. [9] Cold Regions Research and Engineering Laboratory, Hanover, New Hampshire 03755, USA. Correspondence and requests for materials should be addressed to M.J.H. (email: mhoffman@lanl.gov).

I n the ablation zone of the Greenland Ice Sheet (GrIS), the drainage of surface melt to the ice sheet bed via moulins and crevasses causes ice flow acceleration every summer[1–4]. The influx of surface melt overwhelms the capacity of the subglacial drainage system, increasing subglacial water pressure and reducing basal traction of the ice sheet, inducing enhanced sliding[5]. However, ice speed subsequently lowers over the summer despite sustained meltwater input[1–4,6], which is generally explained by increasing efficiency of the subglacial drainage system[2,3,6].

This seasonal evolution of subglacial drainage beneath the GrIS is currently interpreted in the context of traditional theory of a two-component subglacial drainage system consisting of distributed and channelized drainage[1–4,6]. Theory suggests that at low subglacial discharge, drainage occurs through inefficient, distributed pathways—such as linked cavities formed in the lee of bedrock bumps as the glacier slides over the bed or pathways eroded into basal sediments—for which increasing water flux leads to increased water pressure and sliding[7–11]. It is thought that when a critical discharge is reached in the distributed drainage system, dissipation of heat within the water flow causes a positive feedback between melting of the ice roof and cavity growth, leading to the formation of discrete, efficient channels incised into the ice above[5,8,9,11,12]. Such channels would then rapidly evacuate water from the distributed drainage system and lower the water pressures over a large region, terminating a sliding event despite sustained meltwater inputs to the drainage system[5,11–13]. We use a model to illustrate the need for an additional type of drainage system, here termed weakly connected (Fig. 1), and that evolution within this system is responsible for previously unexplained seasonal adjustments to ice velocity.

Observations and modelling suggest that the current conceptual model of the subglacial hydrologic system overemphasizes the role of channelization in controlling GrIS subglacial drainage system capacity and water pressure. Depressed ice speeds persist through fall and much of winter[1,6,14,15], which is inconsistent with a timescale of hours to days for channel collapse under the thick GrIS ice[3,6,16]. Additionally, modelling has indicated that gentle surface slopes on the ice sheet should suppress channel formation[16,17], and low water pressures[18] and depressed summer ice speeds[19] characteristic of highly efficient channels are only observed near the ice sheet margin. Therefore, in interior regions there may be unrecognized drainage capacity elsewhere in the system. Similarly, Andrews et al.[4] recently described direct observational evidence that even where channelization occurs, it is unable to explain lowering ice speed during late summer. At the extensively studied drill site FOXX in western Greenland[4,14,20–23] (Fig. 2a), water pressure in moulins feeding subglacial channels showed little change during the latter part of the melt season, yet velocities were observed to decrease over this same time period (Fig. 3a). Simultaneous observations of declining borehole water pressures sampling poorly connected regions of the bed suggested that weak drainage out of these isolated regions could potentially account for the unexpected increasing system efficiency.

Here, we use a subglacial hydrology model (see Methods: Subglacial hydrology model overview) to demonstrate that the observations at FOXX can be explained by gradual evacuation of water from weakly connected, but spatially extensive, areas of the bed. We suggest that these areas exert the dominant control on the large-scale subglacial water pressure and basal resistance felt by the ice sheet, which we show by using the modelled ice effective pressure to force an ice dynamics model[13,24] (see Methods: ice velocity calculations) to reproduce observed summer ice speed changes. Our subglacial hydrology model includes coupled components for distributed drainage, channelized drainage and drainage from weakly connected

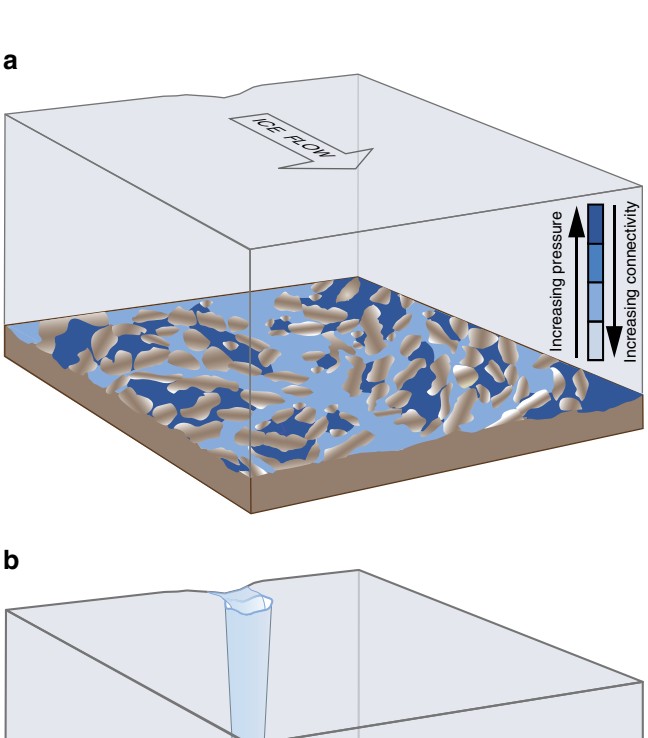

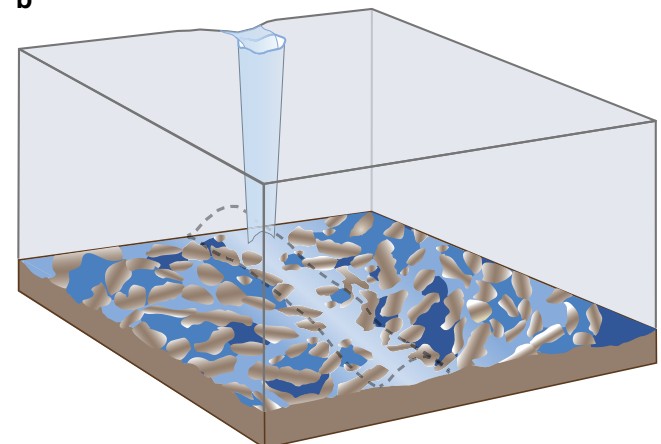

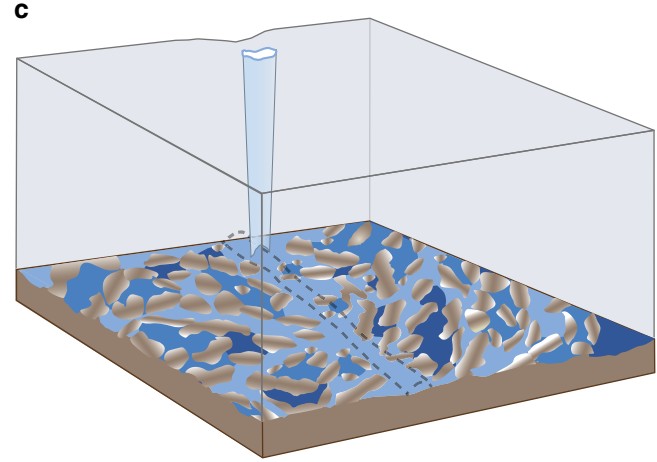

**Figure 1 | Conceptual model of three-component subglacial hydrologic system for the GrIS.** (**a**) Onset of the melt season: a large fraction of the bed is composed of weakly connected cavities at a higher water pressure than the surrounding distributed system. (**b**) Middle of melt season: meltwater draining from the surface through moulins is largely accommodated by the formation of efficient channels (dashed grey outline). Concurrently, some of the weakly connected cavities have leaked water, lowering their water pressure, due to increasing connectivity with the rest of the system initiated during periods of pressurization. (**c**) End of melt season: channels collapse within days after melt inputs cease, but the partially drained weakly connected cavities take months to recharge by basal melting, leaving higher integrated basal traction than before summer began.

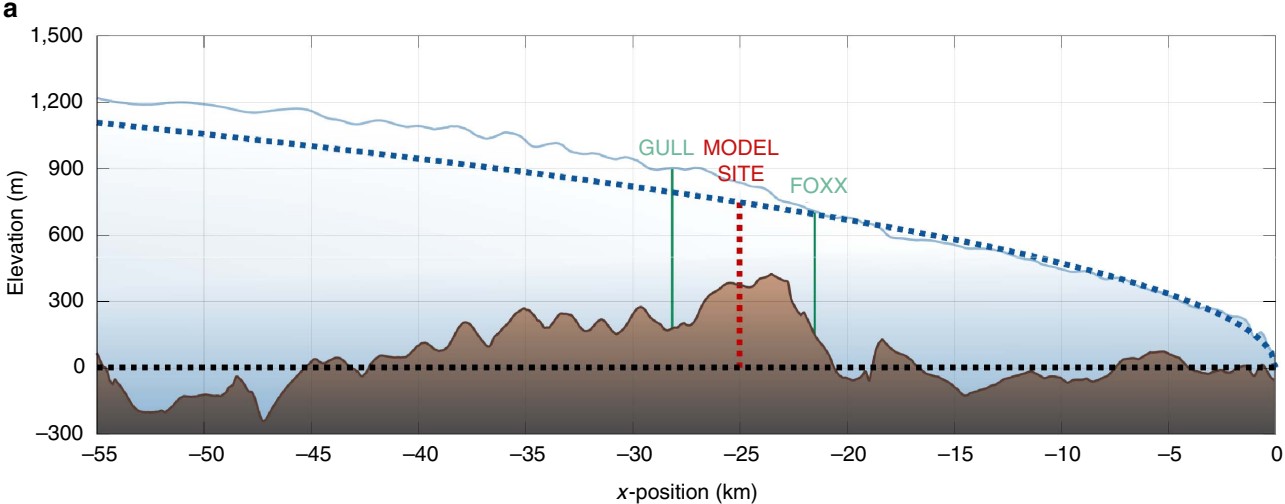

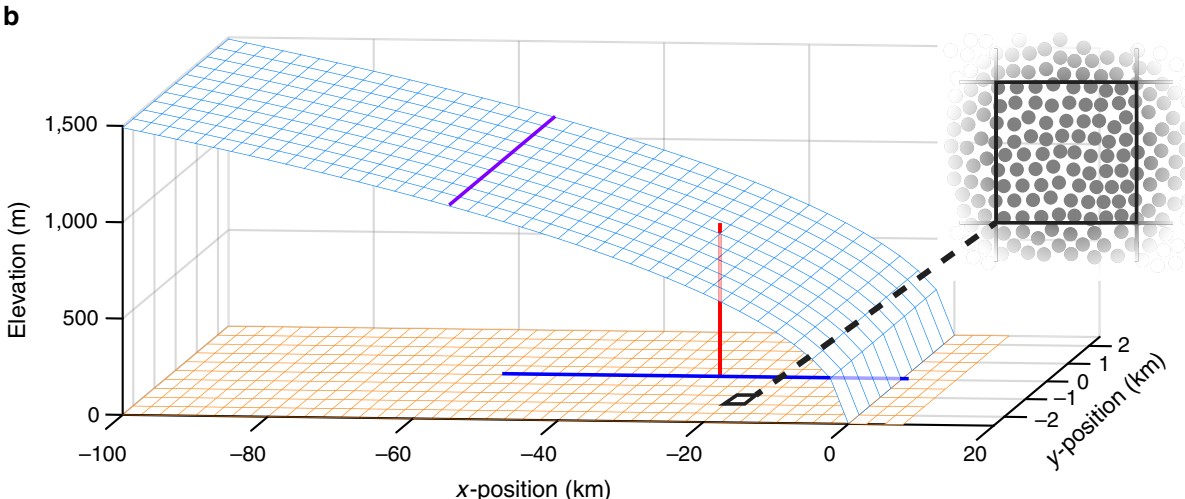

**Figure 2 | Study area and model domain.** (**a**) Ice surface and bed longitudinal profile for GrIS flowline at FOXX study site (thin lines) compared with those for idealized model domain (thick dashed lines). (**b**) Schematic of model domain showing ice surface (light blue) and bed topography (orange). The model study site is indicated by red vertical line. The channel added during the summer simulations is shown by the dark blue line. Purple line on ice sheet surface indicates equilibrium line above which no runoff occurs. Note that the figure is not to scale and the actual grid spacing used in the model is $\Delta x = \Delta y = 200$ m. Inset: schematic of how weakly connected system is represented in the numerical model. Within each grid cell (black box), a fraction of the area, $f_w$, is assumed to be covered by patches of the weakly connected system (dark regions), with the remaining area being composed of the distributed system (light regions). For simplicity, the shape of the weakly connected system is assumed to be many small circular regions, though in reality we expect it to be highly irregular.

regions of the bed (see Methods: weakly connected drainage model). While the former two components have been routinely included in subglacial hydrology system models[11], this is the first application of the latter component. The simulations use an idealized ice geometry based on drill site FOXX and are forced by surface meltwater input into the channelized system based on observed melt rates and by observed ice sheet sliding[14] (Fig. 4a; see Methods: model setup). We conceptualize the weakly connected regions as discrete patches of linked cavities (Fig. 2b), similar to the distributed drainage component (Supplementary Methods), but with a much lower hydraulic connectivity (see Methods: weakly connected drainage model; Supplementary Table 1). Based on observed diurnal and seasonal changes in water pressure in moulins and boreholes[4,20], we assume these patches cover roughly two-thirds of the area of the bed (see Methods: model sensitivity to weakly connected area fraction). There is no through-flow between individual weakly

connected patches; instead, water movement occurs as a 'leaky' exchange with the surrounding distributed system in each grid cell of the model and which is prescribed to become more transmissive over summer.

## Results

**Modelled channelized drainage.** Hydraulic head (a measure of water pressure, see equation (8)) in the modelled channel demonstrates correspondence to hydraulic head measured in a moulin at FOXX (Fig. 4b), which was interpreted as representing the channelized drainage system[4]. Our model results show that an efficient channel remains in approximate equilibrium with melt inputs for the second half of summer (Figs 4 and 5). This is consistent with observations and channel modelling performed by Andrews *et al.*[4] and in contrast to channel modelling performed for locations further inland on the GrIS where flatter

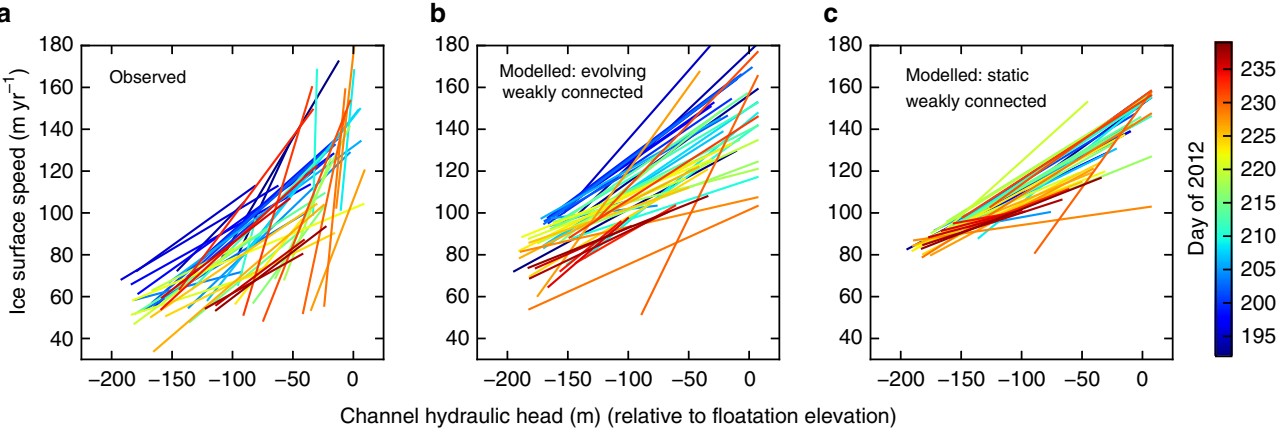

**Figure 3 | Seasonal evolution of the relationship between subglacial water pressure and ice speed.** Each plot shows the minimum and maximum daily values of hydraulic head in the moulin-channel system and ice surface speed for the second half of the 2012 summer. (**a**) Observed relationship showing seasonal hysteresis of lowering ice speed for the same moulin head as summer progresses. Modified from Andrews et al.[4] (**b**) Modelled relationship with increasing permeability of the weakly connected regions of the bed showing similar relationship. (**c**) Modelled relationship from control simulation with static weakly connected system showing lack of coherent seasonal evolution.

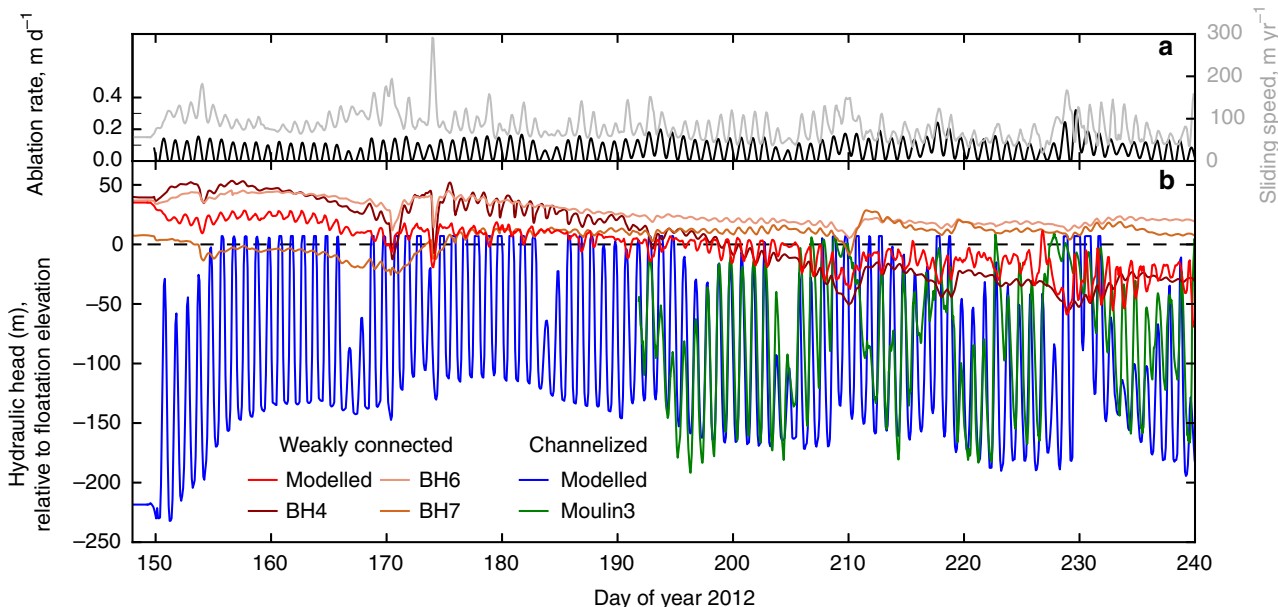

**Figure 4 | Model results for subglacial hydraulic head.** (**a**) Observed melt rate (black) and ice sliding speed (grey) used to force the subglacial hydrology model. (**b**) Modelled and observed subglacial hydraulic head in the weakly connected and channelized systems. Modelled channel hydraulic head (blue) reproduces most features of the measured moulin hydraulic head (green). Modelled hydraulic head in the weakly connected system (red) reproduces most features of the hydraulic head measured in boreholes (pink, orange and maroon). BH = borehole. The datum for both observations and model results is the elevation corresponding to local floatation pressure.

slopes and thicker ice have been proposed to prevent channelization[16,17].

A relatively large initial channel area is required in our model to accommodate surface melt draining to the bed quickly enough for pressures below floatation to develop within a few days, as found in previous models during large pulses of melt[25,26]. Previous modelling studies[16,17,27] have highlighted the unrealistically long time scales required for such large channels to develop. However, our results and those of Andrews et al.[4] demonstrate that if channels are able to grow large enough to accommodate surface melt inputs, they can explain the pressure record observed during the second half of summer. Therefore we consider the possibility that channel formation is preconditioned[26,28]. For example, extensive flooding of the bed

at the start of the melt season from supraglacial lake drainage[3,25,29,30] or other moulin development opens cavity space that facilitates channel formation. This is supported by model results (Fig. 5) showing that channel growth is restricted until the cavity space (represented as water layer thickness) in the distributed system has grown to its maximum seasonal value. These results are consistent with studies suggesting an important role of distributed drainage in developing drainage efficiency[13,17]. Alternatively, year to year persistent moulin locations[31] may facilitate repeated occupation by channels of the same locations and rapid channel growth through cumulative erosion of basal sediments creating preferential flow pathways[26,28,32]. Finally, the prescription of a single channel in our model rather than an anastomosing network of channels likely hinders our model's

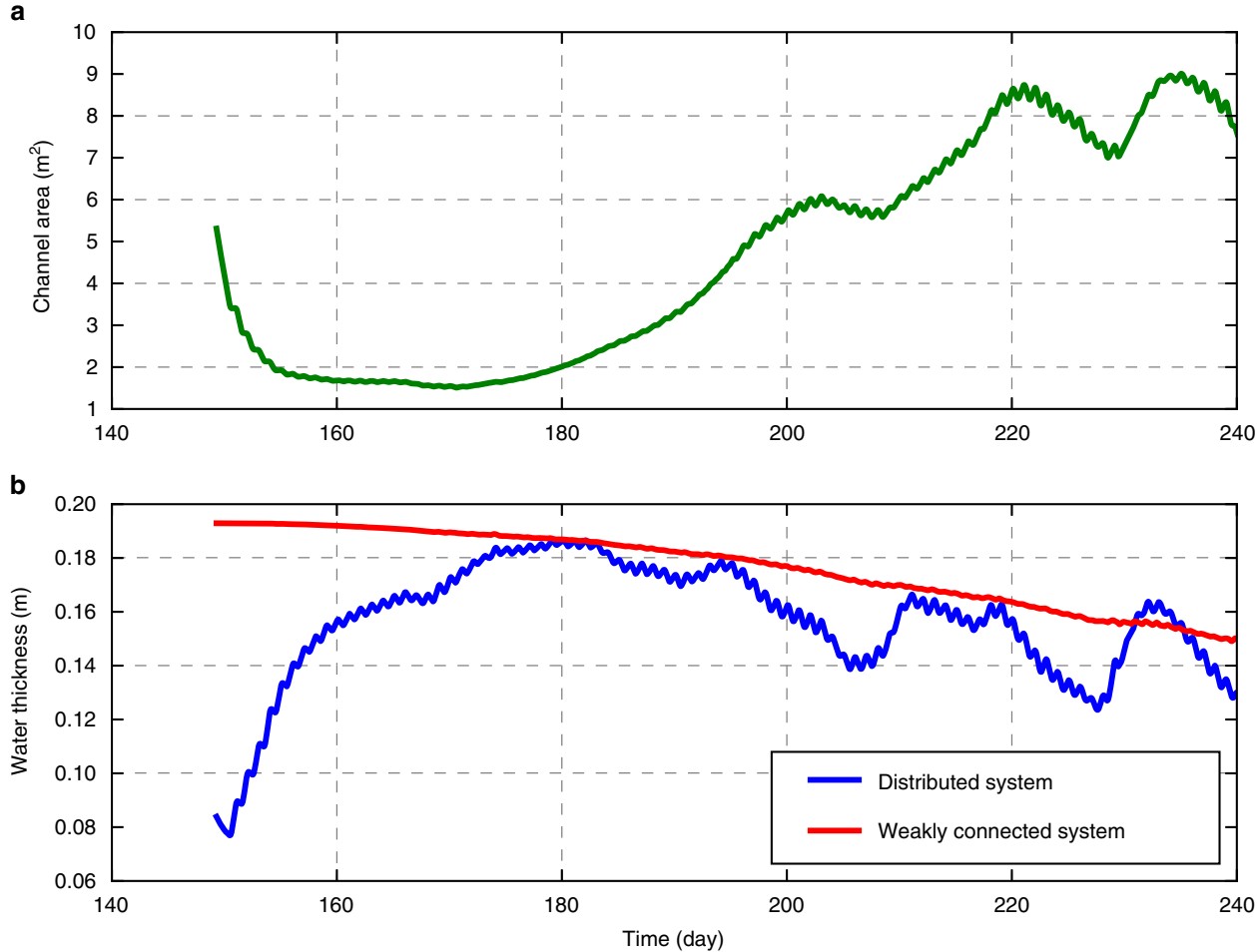

**Figure 5 | Modelled subglacial channel area and water layer thicknesses at study site.** (**a**) Modelled channel area. (**b**) Modelled water layer thickness in distributed (blue) and weakly connected (red) systems.

ability to realistically grow channels while they are small, as the most dominant channels in a network grow in large part by capturing drainage from smaller channels[5,26,33].

**Modelled weakly connected drainage.** Similar to the results for channelized drainage, the modelled hydraulic head in the weakly connected system reproduces the seasonal changes in hydraulic head observed in the boreholes (Fig. 4b). In the model and in measured borehole 4 (and to a lesser extent, borehole 6, see Andrews et al.[4] and Fig. 3d), hydraulic head at the beginning of summer is above that corresponding to ice overburden pressure and then gradually decreases over the course of the season. The trends in hydraulic head in the modelled and measured weakly connected system follow those in the modelled channel during the first part of summer as the channel grows to equilibrium, supporting our hypothesis that dropping summer borehole pressures are caused by slow, down-gradient leakage towards well-connected portions of the drainage system. After the channel reaches its equilibrium size (day ~200–220; Fig. 5), hydraulic head in the weakly connected system continues to drop due to the enhanced connectivity between the weakly connected system and the rest of the drainage system.

This seasonal pattern is overlain by short-term variations in the weakly connected system that contrast with behaviour in the well-connected drainage system. On almost all days, modelled hydraulic head in the weakly connected system is out of phase

with channel pressure, with diurnal amplitudes of a few percent of overburden, matching borehole observations. This out of phase behaviour, observed for isolated or weakly connected boreholes on both the GrIS and mountain glaciers[4,20,34–37], has been explained as the transfer of normal stress from hydraulically well-connected regions of the bed[20,34,36,37], which at our site occurs over kilometers[20]. Because our model does not include diurnal variations in normal stress transfer, our results indicate that diurnal variations in cavity opening rate associated with changes in ice sliding are a possible alternative or additional mechanism for inducing pressure variations that are out of phase with the well-connected drainage system[4,13,37]. We note that the modelled diurnal amplitude of these pressure variations grows unrealistically near the end of summer, which may indicate that a more sophisticated description for cavity opening than our simple linear parameterization (equation (2)) is required to explain all of the observations.

**Modelled ice velocity.** Having validated model water pressure results, we consider implications of the inclusion of the weakly connected drainage component on ice dynamics. The ice dynamics model (see Methods: Ice velocity calculations) reproduces the basic features of the ice velocity observations: while higher channel pressure results in higher ice speed on each day, there is a drop in ice speed for the same channel pressure as the summer progresses (Fig. 3a,b). To eliminate the possibility

that the seasonal hysteresis in the relationship between channel water pressure and ice velocity could be caused by the channelized and distributed components evolving in capacity as the summer progresses, we perform an additional control simulation where those two components are free to evolve as in the first simulation, but the weakly connected component is static with a prescribed, fixed water pressure. In this control simulation, there is limited seasonal evolution (Fig. 3c) supporting our conclusion that the weakly connected system is controlling the late summer slowdown and that inclusion of the weakly connected system is necessary to reproduce the observed changes in ice dynamics.

It is notable that the observations and both model versions exhibit variations in the slope of the lines defining the minimum and maximum channel hydraulic head and ice surface speed on each day (Fig. 3). These varying slopes represent differences in the sensitivity of sliding to changes in effective pressure as effective pressure changes. This changing relation is expected from theory[38–40] and observations[4,41], and we confirm that both models share a similar sensitivity with the observations (see Methods: ice velocity calculations; Supplementary Fig. 1). Thus, the varying slope of the lines in Fig. 3a is expected from having a different range of effective pressure on each day and is not associated with evolution within the weakly connected system.

In contrast, evolution within the weakly connected system is required to explain the lowering of the lines in Fig. 3a as the summer progresses (the downward propagating 'rainbow' pattern in the plot). This downward trend represents seasonal changes in the relationship between moulin water pressure and sliding—the same water pressure induces less sliding later in the season. To quantify this behaviour, we calculate the Pearson product–moment correlation coefficient between the observations and model results for the intercept of each line with hydraulic head of $-75\,\mathrm{m}$ (this value chosen as approximately the centre of the range of hydraulic head values). The seasonal evolution in the model with evolving weakly connected system has a significant positive correlation with the observations ($r = +0.36$, $P = 0.01$), while the model with the static weakly connected system is not significantly correlated with the observations ($r = +0.12$, $P = 0.43$).

Because some observations on mountain glaciers have shown that 'isolated' boreholes can become connected during periods of high water pressure in the active drainage system[34,35], we consider an alternative hypothesis that it is changes to the areal extent of the weakly connected system, and not changes in its connectivity, that cause declining ice speed. However, we find that such a parameterization primarily affects the diurnal range of ice speed and results in minimal changes to ice speed at the seasonal scale (see Methods: model sensitivity to weakly connected area fraction; Supplementary Figs 2 and 3). While we cannot rule out the possibility that the area fraction of the weakly connected system changes modestly during summer, our model results indicate it is not the primary mechanism causing ice speed to drop.

**Recharge time scale**. Tedstone et al.[42] find annual ice motion is more strongly correlated to summer melt volume from the previous 1 to 4 years than summer melt from the corresponding year, and they suggest that this multiyear response is due to 'gradual net drainage of water stored in unchannelized regions'. Our model predicts that water layer thickness in the distributed system remains elevated over the entire summer relative to its pre-summer value, while the water layer thickness in the weakly connected system gradually lowers during summer (Fig. 5). Thus, based on our results, net summer drainage only occurs from the weakly connected system, suggesting that it is these regions that

are controlling the multiyear changes in ice motion. A simple calculation of the time scale of recharge of the weakly connected system made using the basal melt rate and assuming no water drains out suggests that it would take the weakly connected system $\sim 2.0$ years to return to its original water thickness. This estimate is a minimum value because water would continue to drain out as melt refills the system unless exchange completely stopped, which is unlikely. If exchange between the weakly connected and distributed systems is included, assuming the hydraulic gradient at the end of summer and that the permeability immediately returns to its winter value at the end of summer, the recharge time scale increases to 5.1 years. Of course, this is an illustrative time scale because the extent to which weakly connected cavities drain during summer is expected to have significant variation, and not all cavities demonstrate significant drainage[4,20]. Based on these estimates, varying degrees of drainage from the weakly connected system during each summer provide a plausible explanation for the multiyear self-regulation of annual velocity hypothesized by Tedstone et al.[42]

**Conceptual model**. While subglacial drainage has generally been categorized into distributed and channelized components[5,10–12], there is ample evidence that large fractions of the beds of glaciers are 'inactive' and largely disconnected from the pathways directly conveying surface melt that has drained to the bed. These regions are identified by boreholes that drain slowly when reaching the bed and exhibit diurnal water pressure variations that are out of phase with meltwater input and ice speed[4,20,34–37,43]. In many areas, these inactive regions cover large fractions of the bed; Hodge[44] estimated 90% of the bed of South Cascade Glacier in Washington, USA, was hydraulically isolated. This behaviour was observed at all three of the FOXX boreholes, as well as three boreholes at an additional study site[4,20], and boreholes only 30 m apart showed no obvious connection and large hydraulic gradients[20], suggesting that weakly connected drainage likely covers most of the GrIS bed in this region. The implications of large fractions of the bed being relatively isolated at the ice sheet scale has not been previously considered.

Our modelling results confirm that these isolated or weakly connected regions of the bed can play an important role where present by virtue of their large area fraction; modest changes in effective pressure in these extensive regions will have a large impact on basal traction. Importantly, ice dynamics respond to the integrated basal traction over both well-connected and poorly connected regions of the bed, and poorly connected regions therefore moderate active drainage regions[40,43]. The heterogeneous nature of basal drainage (active regions interwoven with weakly connected regions on length scales of metres to tens of metres) is smoothed out because ice dynamics responds to basal traction at the length scale of a few ice thicknesses[45–47] ($> \sim \mathrm{km}$ for GrIS). Both observations and our modelling efforts indicate that, despite the apparent isolation of these areas, they are not entirely static. Often, water pressure changes occur in these areas when water pressure in the active moulin-connected system increases or inactive patches switch to active behaviour[4,34–36].

Based on this new understanding, we propose a three-component conceptual model for subglacial hydrology that adds a 'weakly connected system' to the traditional distributed and channelized forms of subglacial drainage (Fig. 1). We find this third component necessary to model the ice sheet velocity response properly. Meltwater input of sufficient magnitude drives the formation of channels which are the primary control on overall system efficiency. Channels, as inherently linear features,

have little direct effect on ice sheet basal traction, but indirectly control basal traction by acting to lower subglacial water pressure within the spatially expansive distributed system. The strong connectivity between these two components acts to couple the channel-driven drainage state to the ice dynamics. The new qualitative behaviour we are proposing occurs in regions of the distributed system that are so weakly connected to the rest of the bed that they act as a regulator on the active system[43], meaning changes there affect integrated basal traction.

We have envisaged the weakly connected system being composed of cavities formed in the lee of bedrock bumps or clasts in lodged till. The low hydraulic connectivity that differentiates these regions from the rest of the distributed system may be due to local bed geometry causing smaller cavity orifices or the presence of low hydraulic conductivity till. Indeed, extensive subglacial till has been identified at FOXX[21], and stick-slip ice motion observed seismically is associated with spatially differing connectivity of till[23]. More generally, both modelling[48,49] and observations[21,50] have argued for the existence of significant amounts of till underlying marginal regions of the GrIS. We find an alternative formulation of the weakly connected system being composed entirely of till yields similar results (Supplementary Methods, Supplementary Table 2 and Supplementary Fig. 4), and we hypothesize that the weakly connected system is composed of a mixture of cavities and till. We emphasize that the weakly connected system, as conceptualized here, is a subset of distributed drainage with extremely low hydraulic conductivity, and an alternative modelling approach would be to model a distributed system with spatially variable conductivity at high grid resolution. However, we propose it as a separate drainage component due to the qualitatively different behaviour observed there; a broad parameterization of weakly connected drainage is consistent with the parameterized descriptions typically employed for distributed and channelized drainage[10–12].

## Discussion

The impact of weakly connected drainage on the seasonal evolution of hydrodynamic coupling of the GrIS explains apparent contradictions in a less complex conceptual model. First, our model can explain recent observations[4] of water pressure in moulins that show subglacial channels quickly equilibrate with meltwater input while ice velocity continues to decline over the summer. Second, the time scale for re-equilibration of low pressure conduits after the cessation of melt is hours to days[3,6,16], yet the impact of summer melt on GrIS ice dynamics is clearly sustained through most of the winter and may extend for multiple years[6,15,42]. This points to a slow-reacting component of the basal drainage system. The gradual dewatering of the weakly connected system would take longer to recharge than the other systems[42], which is confirmed by a recharge time scale of years in our model.

While the annual cycle of dewatering and recharge of weakly connected regions of the bed suggests a resilience of GrIS ice dynamics to increasing melt[42], the likely importance of erodible till for governing the connections of the weakly connected system to more active regions of the drainage system could allow rapid changes in flow resistance[21]. Furthermore, the spacing of channels, and the moulins that feed them, may control the extent to which weakly connected areas are 'tapped' during the melt season. Future work should attempt to improve direct observations of these weakly connected regions of the bed and improve their description in models to better constrain future changes of hydrodynamic coupling of the GrIS and its effects on ice sheet mass balance.

## Methods

**Subglacial hydrology model overview.** The subglacial hydrology model is based on the model described by Hoffman and Price[13] for coupled distributed and channelized drainage. The model takes a continuum approach to subglacial drainage that captures the bulk behaviour of the subglacial drainage system at spatial scales relevant to ice dynamics ($\sim 10^2$ m) rather than attempting to model every basal conduit explicitly. The three modes of drainage, distributed, channelized and weakly connected, are modelled as separate components, each with appropriate physics, that are coupled together by exchanges of water driven by gradients in hydropotential between the systems. Distributed drainage is modelled as a continuous macroporous sheet on two dimensions on the primary model grid, while channelized drainage is represented by a single one-dimensional channel located along a line of grid cell edges of the primary model grid taking up no area within the primary model grid (Fig. 2b). Summaries of the previously described[13], distributed and channelized drainage components are included in the Supplementary Methods for completeness. The weakly connected cavity system is represented as a subgrid element within the distributed system (Fig. 2b, inset). Within each grid cell of the model, a fraction of the bed, $f_w$, is assumed to be covered by the weakly connected cavity system, with the remaining fraction, $1 - f_w$, covered by the through-flowing distributed drainage system. Surface melt draining from the surface is delivered to the channelized system, which can exchange water with the distributed system. The distributed system can exchange water with both the channelized system and the weakly connected system, while the weakly connected system and the channelized system have no direct exchange. In all three systems, conduit space is assumed to be entirely filled with water at all times, a common approach in subglacial hydrology models (c.f. Schoof et al.[51]). All model parameters are listed in Supplementary Table 1 and are spatially uniform unless otherwise mentioned.

**Weakly connected drainage model.** The weakly connected component is implemented as a discontinuous reservoir that can exchange water with the surrounding distributed drainage system within each grid cell (Fig. 2b, inset). The weakly connected areas are prescribed to cover two-thirds of each grid cell, with the remainder covered by the distributed system. This fraction is chosen to capture the fact that all boreholes observed at drill site FOXX[4,20] exhibited characteristics of hydraulic isolation from the active drainage system over most of the observing period while avoiding being overly prescriptive with the new component.

Using a macroporous sheet continuum formulation for the weakly connected system analogous to that used for the distributed system (Supplementary Methods), mass conservation of the water thickness in the weakly connected component ($h_w$) is described by the balance between locally generated melt ($m_w$) and exchange of water with the sheet ($\gamma_w$),

$$\frac{\partial h_w}{\partial t} = \frac{m_w}{\rho_w} + \frac{\gamma_w}{A_w}, \qquad (1)$$

where $\rho_w$ is the density of water, $A_w$ is the area of the weakly connected system within each grid cell, defined as $f_w \Delta x \Delta y$.

Evolution of cavity space within the weakly connected component is the same as for the distributed system, a balance of cavity opening by sliding of the ice over bedrock bumps and close by creep of their ice roof:

$$\frac{\partial h_w}{\partial t} = |\mathbf{u_b}| \frac{h_r - h_w}{l_r} - \frac{2A}{27} h_w N_w^3, \qquad (2)$$

where $\mathbf{u_b}$ is the sliding velocity, $A$ is the temperature-dependent rate factor for ice deformation, $N_w$ is the effective pressure in the weakly connected system, and $h_r$ and $l_r$ are parameters describing the height and wavelength, respectively, of bumps on the bed. The value for $A$ for the basal layer (Supplementary Table 1) is approximated from results of borehole temperature and deformation observations and ice flow modelling performed by Ryser et al.[14] where the basal layer is temperate and an enhancement factor of 2.5–4 (taken here as 2.7) is appropriate for ice from the late Wisconsin found at this depth.

Energy for local melting within the weakly connected component, $m_w$, also matches that for the distributed system:

$$m_w L = G - \mathbf{u_b} \cdot \tau_\mathbf{b}, \qquad (3)$$

where $G$ is the geothermal heat flux, $\tau_\mathbf{b}$ is the basal traction vector and $L$ is the latent heat of fusion of water.

For all components, hydraulic potential is defined as

$$\phi_{\{d,c,w\}} = \rho_w g z_b + p_{\{d,c,w\}} = \rho_w g z_b + p_{ice} - N_{\{d,c,w\}}, \qquad (4)$$

where the subscript $_{\{d,c,w\}}$ indicates one of the distributed, channel, or weakly connected systems, respectively. $g$ is the acceleration due to gravity (m$^2$ s$^{-1}$), $z_b$ is the bed elevation (m), $p$ is water pressure (Pa) and $p_{ice}$ is the ice overburden pressure (Pa). For the distributed and channel systems the ice overburden pressure is calculated using a hydrostatic assumption, $p_{ice} = \rho_i g H$, where $\rho_i$ is the ice density (kg m$^{-3}$) and $H$ is the ice thickness (m). To allow the weakly connected system to attain water pressure greater than floatation, as observed, we include a simple representation for normal stress transfer[34,36] in the weakly connected system by

increasing the ice overburden pressure according to

$$p_{\text{ice}_w} = \rho_i g H (1 + r_w), \qquad (5)$$

We set $r_w$ to a constant value of 0.13, because this corresponds to the maximum value of borehole water pressure observed at our study site[20], and thus suggests a limiting $p_{\text{ice}_w}$ value when $N_w = 0$. Note that this simple representation does not include temporal variations in normal stress transfer.

Similar to the coupling between the channel and distributed system (Supplementary Methods), the weakly connected component is coupled to the distributed system by calculating a flux, $\gamma_w$, between the surrounding distributed system and the weakly connected system within each grid cell based on a Darcy flow law:

$$\gamma_w = -\frac{k_{0w} h_w^3}{\eta_w} \frac{\phi_w - \phi_d}{\Delta s} P_w, \qquad (6)$$

where $\phi_w$ and $\phi_d$ are the hydraulic potential in the weakly connected and distributed systems, respectively, $k_{0w}$ is the permeability between the weakly connected and distributed systems, $P_w$ represents the perimeter between the two systems within each grid cell and $s$ is a characteristic spacing between the two systems.

While in reality the boundary between the two systems is likely to be complex, for simplicity in our parameterization we assume a simple geometry for the weakly connected patches within each grid cell for the purposes of generating reasonable values for $P_w$ and $\Delta s$. Specifically, we assume each weakly connected patch is a circle of radius 10 m (based on observations of differing connectivity on that scale in GrIS and elsewhere[4,34,35,52]) and set $\Delta s = 10$ m. For the chosen values of the fractional area of the weakly connected system ($f_w = 0.67$) and grid spacing ($\Delta x = \Delta y = 200$ m) we can calculate $P_w \approx 5,355$ m from basic geometry. While other choices could be made here, these details are not important from a practical standpoint without detailed knowledge of the bed, as these parameters, along with $k_{0w}$, are free variables in equation (6). Essentially, equation (6) parameterizes the exchange of water between the two systems primarily as a function of the difference in hydraulic potential and a permeability constant. As with the exchange between the distributed and channelized components ($\gamma_c$; Supplementary Methods), the exchange between the weakly connected and distributed systems can occur in either direction, with water always moving from the component with higher hydraulic potential to that with lower.

The permeability between the weakly connected and distributed systems, $k_{0w}$, is many times smaller than the permeability within the distributed system itself, $k_0$ (by ∼9 orders of magnitude in our model configuration; Supplementary Table 1), defining the weakly connected nature of this new component. To parameterize increases in connectivity between isolated regions of the bed and the active drainage system that are inferred to occur during summer, we allow $k_{0w}$ to increase over the course of the summer by

$$k_{0w} = k_{0w_{\text{winter}}} + k_{\text{rate}}(t - t_s), \qquad (7)$$

where $k_{0w_{\text{winter}}}$ is a base value during winter, and $k_{\text{rate}}$ is a rate at which the permeability increases beginning at the start of summer, $t_s$. We choose the value of $k_{\text{rate}}$ to reproduce observations of borehole water pressure (Fig. 4b). With our parameter choices, $k_{0w}$ increases by a factor of ∼70× over summer; the end of summer value of $k_{0w}$ remains ∼8 orders of magnitude smaller than the permeability constant for the distributed system. While it changes in time, the permeability is spatially uniform. This is a simple parameterization for the increased connectivity during summer in some boreholes observed at site FOXX[4,20] and at mountain glaciers[34,35]. It should be highlighted that there currently is little observational or physical basis by which to construct a governing relation for how permeability may evolve during periods of meltwater input, and improving on the simple linear relationship used here is a critical area for further research. Having explored a number of possible, more complicated ad hoc relations, we found that any relation that caused $k_{0w}$ to increase substantially during summer generated similar qualitative behaviour.

Acknowledging that the weakly connected system may be a subset of distributed drainage but with very low permeability, an alternative implementation would be to directly model a distributed system with spatially varying permeability, avoiding the need for a new component. From a practical perspective, the subgrid parameterization of a third component used here is advantageous for two reasons. First, because GPS and satellite measurements of ice surface velocity indicate smooth velocity fields, it can be inferred that spatial variability in effective pressure and associated drainage conditions at the bed is at the length scale of an ice thickness or less. Representing such spatial heterogeneity at the grid scale would require a very fine grid, while the subgrid parameterization allows the weakly connected system to be represented at a coarser resolution, making this approach transferrable to large-scale ice sheet models. Second, explicitly modelling variable permeability of the distributed system would require additional assumptions and parameters about the spatial distribution of these variations.

Our parsimonious parameterization of the weakly connected system leaves room for additional complexity as empirical knowledge of the system increases. We have assumed in our model that all parts of the weakly connected system have changing permeability during summer (equation (7)), while borehole observations suggest that only some weakly connected cavities become 'leakier' during summer[4]. Similarly, some studies have observed weakly connected cavities becoming fully

'connected' during periods of high water pressure in the surrounding drainage system[34,35], which would correspond to temporal changes in our $f_w$ parameter that we have kept steady in time (see Methods: model sensitivity to weakly connected area fraction for additional discussion). Additionally, the bedrock geometry for the weakly connected system may have different characteristics ($h_r$, $l_r$) than for the distributed system. Though we acknowledge the inherent complexity of the subglacial system, we have chosen the simplest formulation that includes the dominant processes inferred to occur. Our parameterization of the weakly connected system is meant to represent the mean conditions of the bed and thus will not necessarily directly reflect unique measurements from specific boreholes.

**Model setup.** We generate a simplified model domain that is consistent with the basic geometry of our study site (Fig. 2). The domain represents a 100 km long sector of the GrIS from margin to lower accumulation zone, with our study site located 25 km inland from the ice margin. The domain is 5 km wide with periodic lateral boundaries to approximate the typical width of a supra- and sub-glacial catchment in this region of GrIS (estimated moulin density in this region is 0.2–0.25 km$^{-1}$ (refs 1,53)). The ice sheet geometry is a flat bed with a 'plastic' glacier shape[51,54] assuming a constant basal shear stress of $10^5$ Pa. This geometry provides an idealized but consistent setting with our study site (Fig. 2 and Supplementary Table 3). The subglacial hydrology model is applied for the entire domain, but model results are primarily analysed at the study site location (Fig. 2 and Supplementary Table 3); the larger domain is used only to generate realistic far-field constraints on the model solution at the study site. There is a zero subglacial water flux boundary condition for the distributed system at $x = -100$ km, and water pressure in the distributed system is assumed to be at atmospheric pressure at the ice sheet margin, $x = 0$ km.

We choose this simplified geometry over the more complex geometry of the field site to simplify our interpretation of model results, given uncertainty in the true ice sheet geometry. Previous work in our study area has shown that complex topography can affect sliding and deformation over short distances[14] and that 'active' and 'passive' subglacial regions might be affected by bed topography[20]. However, the two-dimensional basal topography of our study region is only known approximately. Only a handful of flight lines of airborne ice thickness measurements exist in close proximity to our study area, and for those that do, differences in ice thickness at flightline crossover points exceed 100 m in some locations. Additionally, the commonly used BedMachine product[55], which uses mass conservation constraints to improve estimates of ice thickness between radar flight lines, does not cover this area. Previous subglacial hydrologic modelling has shown that valleys in the bedrock topography can concentrate subglacial drainage[33], so the true bedrock topography is likely to play an important role in defining the large-scale drainage network and the initial rate of subglacial channelization. However, the location of moulins that input water to the basal drainage system is the key control on channel initiation and stability[26,28,33]. Because both our observations and simulations are focused on a site at a moulin, we expect the simplified model geometry to have little effect on the local subglacial drainage conditions at the study site.

To generate a model initial condition for the summer simulations, we spin up the coupled distributed and weakly connected systems to steady state, assuming no channelized drainage, to represent late winter conditions. We use the winter ice sliding speed to force the model and apply no meltwater forcing. Model parameters $h_r$, $l_r$ and $k_0$ (Supplementary Table 1) are chosen to yield a hydraulic head in the distributed system at our study site of about 200 m below local floatation elevation, and $k_{0w_{\text{winter}}}$ is chosen to yield a hydraulic head in the weakly connected system of about 40 m above local floatation elevation.

**Model forcing data.** The subglacial hydrology model uses time series of two forcing fields, both of which are based on observations in the vicinity of the FOXX drill site. Surface melt input to the subglacial drainage system ($\omega$ in Supplementary Equation (5)) represents the volume flux of water drained to the bed through moulins and crevasses, and is the primary source of new water to the subglacial drainage system during summer (basal melting being of much smaller magnitude). Ice sliding speed ($\mathbf{u_b}$ in equation (2) and Supplementary Equation (2)) controls the rate at which new cavity space opens in the distributed and weakly connected systems, and, less importantly, the frictional melt rate (equation (3) and Supplementary Equation (4)).

We base the melt forcing (Fig. 4a) off of 6-h average ablation rates[4] measured using a pressure transducer installed below the surface connected to a surface reservoir[56]. Because the measured ablation rates do not have the precision necessary to directly generate a time-series with the required time step of 2-h, and the 6-h average rates reduce the amplitude of the diurnal cycle, we generate an idealized diurnal cycle by fitting a sine curve to the 6-h time-series where we adjust the amplitude on each day to maintain the measured 6-h average. The ablation sensor began to fail after the large melt event on days 229–230, so we estimate the ablation rate for days 231–240 using a linear fit between mean daily air temperature and ablation rate for the 10 days prior to the large melt event (Pearson correlation coefficient $r = 0.702$, $P < 0.01$). We scale the point ablation rate by the width of our model domain and apply it as a linear source term along the length of the channel model ($\omega$). As discrete moulins can result in the generation of kinematic waves within our channel model, this linear distribution of melt input improves model

stability, while reproducing synchronous changes in moulin water level as observed by Andrews et al.[4] We apply a linear lapse rate such that runoff is zero at the equilibrium line altitude of 1,100 m. This provides plausible runoff rates for the entire domain, recognizing that model results will be most sensitive to runoff at the location of our study site which are well constrained. Though supraglacial storage on the GrIS is known to change over the melt season[57], our melt forcing ignores supraglacial and englacial routing and storage processes that affect the timing and magnitude of melt delivery to the bed[58,59], and is meant to be an idealized representation of the diurnal and seasonal variations in melt forcing at our study site.

The ice sliding speed forcing (Fig. 4a) comes from the results of Ryser et al.[14] for site FOXX, which subtracts borehole-derived measurements of ice internal deformation from Global Positioning System-derived measurements of ice surface velocity to calculate basal slip. Four gaps of about a day are filled by averaging the diurnal cycles on either side of the gap and smoothing the edges to avoid any sharp transitions. Basal slip is 73% of motion during winter and, though speeds vary dramatically over summer, the contribution of deformation is roughly constant[14]. We apply this sliding forcing uniformly over our entire model domain. Though ice speed is not spatially uniform in reality, this simplifying assumption will not directly affect our conclusions as we only analyse the model results at our study site. The magnitude of basal traction, $|\tau_b|$ is held constant at $10^5$ Pa for the entire simulation; though it should vary in reality, we do not have the information to provide a more accurate value, and, in any case, the results are not sensitive to this choice as it only affects frictional melting, which is orders of magnitude smaller than summer surface melt input.

**Summer subglacial hydrology simulations.** Using this spun-up state as an initial condition, we model the 2012 summer for days 150 (onset of summer speedup in the GPS record) to 250 (end of GPS and melt forcing observation time-series). For the summer simulation we add a single active channel along the centreline of the domain. While this precludes the formation of a network of channels (c.f. refs 26,33), we expect this approach to resolve the dominant channel effects near our study site, as moulin location strongly controls channel nucleation[28,33]. The channel initial condition is an area of 10.0 m² at the margin decreasing linearly to zero at 55 km inland, which yields an area of 5.45 m² at the study site. This value was chosen to allow water pressure to drop below floatation and diurnal variations to develop within days of melt onset, as seen in the ice velocity record. The channel roughness parameter $F$ is chosen to yield diurnal hydraulic head minima in late summer comparable to that measured in moulins[4] and is broadly consistent with previous modelling efforts.

**Hydraulic head observations.** We compare model results to hydraulic head measured at site FOXX[4,20] in 2012, where hydraulic head, $d$, relates to hydropotential as

$$d = \phi/(\rho_w g), \qquad (8)$$

Modelled hydraulic head in the channelized system is compared with hydraulic head measurements in moulin 3 and modelled hydraulic head in the weakly connected system to hydraulic head measurements in boreholes 4, 6 and 7. Because of the simplified model geometry and our desire to make the comparison of our model data general to both FOXX and neighbouring GULL study sites (which both exhibited similar borehole behaviour)[4,20], there is no direct correspondence in ice thickness and bed elevation between the model and the observational study sites (Supplementary Table 3). Therefore, we plot both the model and measured hydraulic head using $d_f$, the elevation corresponding to floatation pressure, as the vertical datum:

$$d_f = (\rho_w g z_b + \rho_i g H)/(\rho_w g), \qquad (9)$$

which is slightly different at each measurement site (see ref. 4, Extended Data Table 1) and in the model (Supplementary Table 3) due to modest differences in bed elevation and ice thickness.

**Ice velocity calculations.** Ice velocity for Fig. 3b,c is modelled using the thermomechanical, three-dimensional, first-order Stokes approximation[60–63] momentum balance solver in the Community Ice Sheet Model v2.0[24] as described by Hoffman and Price[13]. The magnitude of basal traction, $\tau_b$, is defined by a physically based basal friction law for sliding over hard beds that allows for cavitation and bounded basal drag[39,64],

$$\tau_b = \tau_b^{\circ} + C\left(\frac{u_b}{u_b + N^n \Lambda}\right)^{1/n} N, \quad \Lambda = \frac{\lambda_{max} A}{m_{max}}, \qquad (10)$$

where $C$ is a Coulomb friction constant, $\lambda_{max}$ and $m_{max}$ are the wavelength (m) and maximum slope, respectively, of the dominant bedrock bumps, and $n$ is the exponent in Glen's flow law. $\tau_b^{\circ}$ is an addition to the original formulation added here to make basal traction calculated for our simplified model domain more realistic. Because our domain lacks the rough, heterogeneous topography of the real ice sheet where resistance to flow is likely to be concentrated in our study area[20], we use $\tau_b^{\circ}$ to represent these missing 'sticky' spots[40]. We apply a constant value of $\tau_b^{\circ}$

across our study domain (Supplementary Table 1), chosen to reproduce a realistic range of model speeds across the range of effective pressure forcing applied.

With widespread cavitation (i.e., when effective pressure approaches 0 at high water pressure), the friction law becomes a Coulomb friction law of the form (moving $\tau_b^{\circ}$ to the left-hand side to clarify the form)

$$\tau_b - \tau_b^{\circ} = CN. \qquad (11)$$

Alternatively, at large effective pressures (low water pressure) the friction law takes a power law form

$$\tau_b - \tau_b^{\circ} \propto u_b^{1/n}. \qquad (12)$$

The basal traction appropriate for ice dynamics is the integrated basal traction of both connected and isolated regions of the bed[43],

$$\tau_b - \tau_b^{\circ} = \tau_{bd}(1 - f_w) + \tau_{bw} f_w, \qquad (13)$$

where $\tau_{bd}$ and $\tau_{bw}$ are the basal traction in the distributed and weakly connected systems, respectively.

We approximate equation (13) by assuming the effective pressure used in equation (10) is an area-weighted average in each grid cell, $N_{int}$, of the effective pressure in the distributed and weakly connected systems,

$$N_{int} = N_d(1 - f_w) + N_w f_w, \qquad (14)$$

This simplification is justified by the fact that for the effective pressure and parameter values in the simulations, equation (10) remains primarily in the Coulomb friction law regime (equation (11)) where the use of equation (14) yields the appropriate description of basal traction, equation (13). (This approximation becomes increasingly inaccurate as the basal friction law transitions into the form of equation (12) where $\tau_b$ is not directly proportional to $N$).

Two standalone simulations of ice velocity are performed, both forced by effective pressure generated by the summer subglacial hydrology simulations. In the first simulation, equation (14) is calculated from $N_d$ and $N_w$ calculated in the summer subglacial hydrology simulation. This simulation demonstrates the impact of weakly connected drainage on ice dynamics (Fig. 3b). In the second simulation, the $N_w$ field is held steady while $N_d$ comes from the summer subglacial hydrology simulations (Fig. 3c). This is a control simulation that confirms the lack of seasonal hysteresis in the effective pressure–velocity relationship when the evolution of weakly connected drainage is not included in the model. Comparison of the velocity versus pressure relationship from these two simulations eliminates the possibility that the observed seasonal hysteresis is due to changing conditions in the distributed and/or channelized systems.

In both ice velocity simulations, the temperature-dependent rate factor, $A$, is a function of ice temperature[40], and the vertical ice temperature profile in the model is taken equal to that measured at our study site[14,22] using 11 uniform vertical levels. Based on ice deformation calculations from Ryser et al.[14] we apply an enhancement factor to $A$ of 2.7 × within the deepest modelled layer. Ice geometry is held steady for the duration of these summer simulations; ice velocity is simply calculated diagnostically at each time step based on the fixed geometry and changing basal boundary condition, which is a function of our modelled subglacial hydrological evolution. As for the subglacial hydrology model, the domain is periodic in the $y$-direction. Because we are only concerned with the ice speed at the study site, the domain for the ice dynamics calculations is subset to span the area 10 km upstream and downstream of the study site to make the calculations less expensive. At the upstream and downstream boundaries we apply a vertically uniform Dirichlet boundary condition on the velocity field of 100 m a$^{-1}$. We confirm that the velocity solution at the study site is independent of the choice of boundary condition value (as expected when the boundaries are far enough from the study site, $> \sim 4$–10 ice thicknesses)[65].

Parameters in equation (10) (Supplementary Table 1) are tuned to to yield model velocity of the observed magnitude, and the same values are used for both model versions. Specifically, $\tau_b^{\circ}$ and $C$ are varied in combination to approximately match the observed diurnal range of surface speeds. The ratio $\frac{\lambda_{max}}{m_{max}}$ is tuned to achieve the observed variation in the sensitivity of ice speed to effective pressure. This is assessed by an analysis of the slopes of the lines defining the minimum and maximum channel hydraulic head and ice surface speed on each day in Fig. 3. This slope represents the relationship between water pressure and sliding within a single day. A key feature in the observations (Fig. 3a) is a tendency for lines restricted to lower hydraulic head values to have flatter slopes than those at higher hydraulic head values (lines to the left tend to be more horizontal). This behaviour is the result of a reduction in sliding sensitivity to water pressure at low water pressures (high effective pressures). It is predicted by theory and is a key feature of the basal friction law used (equation (10)). At large effective pressure when subglacial cavities are small, sliding is controlled by regelation and enhanced creep[38–40]. This insensitivity of velocity to effective pressure at large effective pressures has been observed in mountain glaciers[41] and can be clearly seen in high temporal resolution data at our study site (see ref. 4; Extended Data Fig. 4b). To ensure the models are calibrated to correctly represent this changing sensitivity of sliding to effective pressure, we calculate a linear regression between the minimum channel hydraulic head and the slope of the channel hydraulic head/ice speed relationship on each day. Restricting the regression to the range of hydraulic head minimum values in common between all three data sets ($<125$ m), we confirm that the observations and both model versions have a similar sensitivity to changes in effective pressure

for the parameter values used (Supplementary Fig. 1). The differing slopes of the lines in Fig. 3c represent this variable sensitivity of sliding to effective pressure and are what cause the modest seasonal-scale changes exhibited by the model with a static weakly connected system.

**Model sensitivity to weakly connected area fraction.** While a complete analysis of the sensitivity of all parameters used in the models is beyond the scope of this study, we assess how the study's main conclusions are affected by the choice of key parameter, $f_w$, the area fraction of the weakly connected system, and the possibility that it could change in time.

In the main text, we present results for $f_w = 0.67$. Because a total of six boreholes at two different drill sites exhibited out-of-phase behaviour for the majority of both summers measured[4,20], we assume the fraction must be large, but we want to avoid being overly prescriptive with the new component. In fact, an upper bound for $f_w$ can be found based on the observations[4,20] that hydraulic head in the moulin system has typical diurnal variations of about 20% of overburden pressure, while diurnal variations in the boreholes are about 2%. Because the observed surface velocity is precisely in-phase with the moulin pressure variations[4] that are driving the system, we can assume that the area-weighted diurnal variations in the distributed system connected to the moulin must be larger than the area-weighted variations in the weakly connected system:

$$0.20(1 - f_w) > 0.02 f_w, \qquad (15)$$

This provides an upper bound of $f_w < 0.91$. Note that if diurnal variations in the moulin are in actuality larger than those across representative regions of the distributed system connected to it (as might be expected for a diffusive pressure wave[32]), then the upper bound for $f_w$ should be smaller than the calculated value.

As anticipated from equation (15), if we prescribe $f_w = 0.90$ within the model (and retuning the parameters in the basal friction law), diurnal variations in ice speed are almost entirely absent (Supplementary Fig. 5). This is because the area-weighted diurnal variations of the weakly connected system roughly cancel out the area-weighted diurnal variations in the distributed system because the two systems are out of phase from each other. The few days with substantial diurnal range in ice speed seen in these model results occur when conditions within the model temporarily shift the phasing of diurnal variations in the weakly connected system. Note that ice speed still drops over the summer when the weakly connected system is allowed to drain (Supplementary Fig. 5b), but overall the results differ markedly from the observations and are largely unphysical.

The lower bound on $f_w$ is less constrained than its upper bound, but based on the weakly connected behaviour of all boreholes in the study area, we assume $f_w = 0.50$ forms a reasonable lower constraint. Model results with that value (Supplementary Fig. 6) are somewhat similar to the baseline value of $f_w = 0.67$ used in the main text (Fig. 3), but the ice speed does not drop as substantially; with the weakly connected system covering a smaller fraction of the bed, changes there have less impact on the integrated basal traction. Noting that the modelled changes in the weakly connected system have been constrained by the borehole observations, we find that $f_w$ values of $\sim 0.60$–$0.80$ can give results that provide a reasonable match to the ice speed measurements, allowing for modest adjustments to the other parameters in the model.

In addition to assessing the baseline value of $f_w$, we also consider the possibility that $f_w$ could change during the summer. Certainly, rather than existing areas of weakly connected drainage becoming more strongly connected to the active drainage system, a reasonable hypothesis would be a change in the area fraction of weakly connected regions. We test this by an additional model run where evolution within the weakly connected system occurs by $f_w$ declining (Supplementary Fig. 2) or increasing (Supplementary Fig. 3) rather than changes to the permeability. These parameterizations do a worse job at reproducing the observed behaviour. Of these two runs, the situation where $f_w$ declines is the more observationally supported change—observations on mountain glaciers have shown that 'isolated' boreholes can become connected during periods of high water pressure in the active drainage system[34,35]. However, in the run where $f_w$ declines, little seasonal evolution in the ice speed occurs (Supplementary Fig. 2b). Instead, the diurnal range in ice speed increases as the summer progresses, an effect that is not seen in the observations. Similarly, the primary effect of increasing $f_w$ during summer (Supplementary Fig. 3b) is a decrease in the diurnal range of ice speed. While we cannot rule out the possibility that the area fraction of the weakly connected system changes during summer, our model results indicate it is not the primary mechanism causing ice speed to drop.

**Data and code availability.** Model code is development code based off of the Community Ice Sheet Model (CISM) version 2.0.4 (http://oceans11.lanl.gov/cism/). Model code, processing scripts, input datasets and model output are all available on request from the corresponding author.

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

## Acknowledgements

This work was supported by a grant to M.J.H. from the Laboratory Directed Research and Development Early Career Research Program (LDRD-ECR) at Los Alamos National Laboratory, Climate Modeling Programs within the U.S. Department of Energy Office of Science, and by the National Science Foundation, under grant ANT-0424589 to the Center for Remote Sensing of Ice Sheets (CReSIS). L.C.A. was supported by an appointment to the NASA Postdoctoral Program at the Goddard Space Flight Center, administered by Universities Space Research Association under contract with NASA, and UTIG Ewing-Worzel and Gale White Graduate Student Fellowships. J.G. was supported by National Science Foundation Division of Earth Sciences (EAR) Postdoctoral Fellowship (No. 0946767). Fieldwork resulting in the presented observations was supported by United States National Science Foundation grants OPP-0908156 and OPP-0909454, Swiss National Science Foundation grant 200021_127197, National Geographic Society grant 9067-12 and NASA Cryospheric Sciences.

## Author contributions

M.J.H. developed the model, designed and ran the experiments, and wrote the manuscript. L.C.A. analysed observational data and contributed to development of the conceptual model. S.A.P. consulted on model implementation, application and interpretation. All authors contributed to analysis and interpretation of observational data and participated in preparation of the manuscript.

## Additional information

**Competing financial interests:** The authors declare no competing financial interests.

**Publisher's note**: 

