## [Peer Review File · Nature Communications]

Reviewers' comments:

Reviewer #1 (Remarks to the Author):

General comments

This modeling study is motivated by data from the western margin of the Greenland ice sheet, where some of the coauthors of the present paper measured water pressure in and under the ice sheet as well as the surface and internal flow velocity of the ice. The conclusion of the observational work was that the seasonal ice-flow dynamics were strongly influenced by the hydraulically unconnected (or poorly connected) fraction of the glacier bed, not just the evolution of an inefficient “distributed” versus efficient “channelized” drainage system. The existence and role of the hydraulically unconnected bed are documented in the alpine glacier literature, but have been hitherto neglected in more recent studies of the hydrologically-forced dynamics of the Greenland ice sheet.

The current paper sets out to test the hypothesis that this poorly connected bed fraction could be responsible for an oft-observed feature of the ice-sheet seasonal cycle of hydrology and dynamics, namely the prolonged period of low flow speeds beginning in late summer. This feature, along with almost any other related to seasonal hydrology, has been attributed to the ubiquitous conceptual model of distributed versus channelized drainage. The authors correctly point out that this conceptual model is oversimplified, and cannot explain all of the observations usually cast under its banner. The importance of the current paper is in presenting a more nuanced but realistic model of subglacial drainage and thus a more powerful framework for interpreting observations from Greenland.

The authors adapt a coupled model of hydrology and ice dynamics that they have used in previous work to examine the interaction of distributed and channelized drainage. Here they partition the distributed system into prescribed connected and unconnected fractions, with a leakage term between the two based on the fluid potential gradient. The channel communicates only with the connected fraction of the distributed system and is fed through a moulin with time-dependent surface melt. The model is applied to a synthetic geometry with characteristics similar to those of the measurement sites and forced by a timeseries of surface melt. Model parameters are constrained by data to the extent possible. The model exhibits qualitatively similar behavior to the observed, namely: (1) a general decrease in ice-flow responsiveness to meltwater input over the latter half of the melt season and (2) out-of-phase diurnal variations between the channel and the poorly-connected bed area, with the latter characterized by high baseline water pressures

and low-amplitude fluctuations. A control simulation that omits the interaction of the poorly connected bed is presented to establish the explanatory power of the poorly connected bed when coupled to the connected fraction of the distributed system. One interesting outcome of the simulations is the suggestion that anti-phase behaviour between the well- and poorly-connected bed areas may be explained by diurnal variations in cavity opening by sliding, in addition to (or instead of) hydromechanical coupling and the transfer of normal stress, as has been previously assumed in the literature. This idea may have been presented in the observational work, but is corroborated here with the model.

Despite its caveats and some of the less-satisfying ad-hoc aspects of the model (see below), this study is novel and important. Though the scientific conclusion of the work has already been put forth in the group's observational papers, this is the first attempt (to my knowledge) to include poorly-connected drainage in an otherwise distributed continuum model of ice-sheet hydrology. The message that observations from Greenland demand a more nuanced understanding than the distributed-versus-channelized conceptual model will elevate the conversation on basal drainage and improve our framework for interpreting hydrologically-forced ice dynamics in Greenland.

Specific comments

1. Though the model presented provides an adequate basis for the claims made in the paper, it still has some unsatisfying elements. The authors have, no doubt, painstakingly tried to deal with the lack of direct observational constraints on model formulation and parameters ubiquitous in any attempt to simulate glacier drainage. In contrast to most other theoretical/modelling studies, the authors are able to draw directly upon data specific to the study site. Still, one is left wondering about the extent to which the results are a function of the chosen formulations or parameters, particularly where the authors state that parameters were chosen to achieve a particular result. Among the more ad-hoc aspects of the model, the authors may want to comment on the sensitivity of results to: (a) A choice of f_w other than $2/3$. This value is justified based on the fact that all of their boreholes tapped into the "weakly connected" drainage system, but how sensitive are the results to, for example, $f_w = 0.5$ or 0.9 ? (b) Constant f_w . Could the authors comment on why they chose to keep f_w constant rather than allowing the bed fraction of connected versus unconnected drainage to evolve? I assume the argument against an evolving f_w is that an additional governing equation (of an unknown form) would be required. (c) The prescribed evolution of the permeability coefficient k_{0w} during the melt season. Does the form of the function matter? Do any of the above (a,b,c) affect the estimated timescales presented at the top of page 6, for example? Finally, the simplified model geometry (Figure 2a) contrasts sharply with reality when it comes to bed geometry. It would be reassuring to see some acknowledgment of, or speculation on, the possible influence of real bed topography on the hydrology. All of these points could be addressed in the Supplementary Material.

2. Success of the coupled simulation versus the control model, Figure 3. This result is central to the claim of the paper: that poorly-connected areas of the bed play a fundamental role in the seasonal dynamics of ice flow. By eye, Figure 2b looks better than Figure 2c in emulating Figure 2a, but not in an overwhelming and obvious way. It would strengthen the authors' central claim if the superior match of 2b were demonstrated in some quantitative metric. It is really the temporal evolution of the slope of the lines that is important here. Perhaps a correlation coefficient between the modeled and observed versions of such a quantity would more convincingly establish that 2b is a better match than 2c.

3. The authors choose to present the weakly connected bed fraction as a third component of the drainage system, in addition to the "distributed" and "channelized" components. In anticipation of this paper's influence and uptake, I would encourage the authors to carefully consider their terminology. Channels are "connected" and efficient by nature, as presented in the paper. The distributed system is inefficient by nature (based on the physics), but it is not clear to me that it must also be fully "connected" as presented in the paper. Have the authors considered casting the distributed system as a mixture of poorly- and well-connected fractions (as the model is currently implemented), rather than contrasting the distributed system (assumed inherently connected) with the weakly connected system? The current justification is given on page 7, lines 187-191, but it could equally be the case that previous work has neglected to acknowledge variable connectivity in the distributed drainage system. A related issue is whether the "weakly connected" drainage system includes truly unconnected bed areas. Though just semantics, these labels should be carefully chosen as they are likely to be propagated in through the literature.

Reviewer #2 (Remarks to the Author):

Review of the article entitled "Greenland subglacial drainage evolution regulated by weakly-connected regions of the bed"

This study investigates the link between the subglacial drainage system of Greenland Ice Sheet and its surface velocities. The novelty of this publication lies in the use of a three component subglacial hydrological model where preceding models were using only two components to describe the subglacial drainage system. The results of the model compared to observations made at a Greenland borehole site show an improvement in the modelling of the subglacial water pressure and hence its influence on the basal velocity of the Ice Sheet.

The Study presented here is clearly written, and sufficiently documented both in terms of data comparison and model presentation. I recommend a publication of this study as is with only a

few minor comments listed below.

Throughout the manuscript, references to the Methods could be more precise to help find relevant informations (equation numbers, section numbers)

Line 15 : "increasing subglacial water pressure" in place of " increasing water pressure"

Line 175: I would omit "of the bed"

Line 202: "...time scale of years in our model." rather than "... time scale in our model of years."

Caption of Figure 2 : Model domain is presented with thick dashed lines and not thin lines as stated. Cyan is probably not the best colour choice for the equilibrium line.

Line 266 . Missing unit for "p_ice"

Line 341 and 342: Hydraulic heads should be expressed in meters, I guess that the percentage refers to percentage of the flotation head

SI Table 3: There are some exponent that should not be there

Response to Reviewers

“Greenland subglacial drainage evolution regulated by weakly-connected regions of the bed” Hoffman et al.

Author responses are added inline in blue, italicized font in the text below.

Reviewer #1 (Remarks to the Author):

General comments

This modeling study is motivated by data from the western margin of the Greenland ice sheet, where some of the coauthors of the present paper measured water pressure in and under the ice sheet as well as the surface and internal flow velocity of the ice. The conclusion of the observational work was that the seasonal ice-flow dynamics were strongly influenced by the hydraulically unconnected (or poorly connected) fraction of the glacier bed, not just the evolution of an inefficient “distributed” versus efficient “channelized” drainage system. The existence and role of the hydraulically unconnected bed are documented in the alpine glacier literature, but have been hitherto neglected in more recent studies of the hydrologically-forced dynamics of the Greenland ice sheet.

The current paper sets out to test the hypothesis that this poorly connected bed fraction could be responsible for an oft-observed feature of the ice-sheet seasonal cycle of hydrology and dynamics, namely the prolonged period of low flow speeds beginning in late summer. This feature, along with almost any other related to seasonal hydrology, has been attributed to the ubiquitous conceptual model of distributed versus channelized drainage. The authors correctly point out that this conceptual model is oversimplified, and cannot explain all of the observations usually cast under its banner. The importance of the current paper is in presenting a more nuanced but realistic model of subglacial drainage and thus a more powerful framework for interpreting observations from Greenland.

The authors adapt a coupled model of hydrology and ice dynamics that they have used in previous work to examine the interaction of distributed and channelized drainage. Here they partition the distributed system into prescribed connected and unconnected fractions, with a leakage term between the two based on the fluid potential gradient. The channel communicates only with the connected fraction of the distributed system and is fed through a moulin with time-dependent surface melt. The model is applied to a synthetic geometry with characteristics similar to those of the measurement sites and forced by a timeseries of surface melt. Model parameters are constrained by data to the extent possible. The model exhibits qualitatively similar behavior to the observed, namely: (1) a general decrease in ice-flow responsiveness to meltwater input over the latter half of the melt season and (2) out-of-phase diurnal variations between the channel and the poorly-connected bed area, with the latter characterized by high baseline water pressures and low-amplitude fluctuations. A control simulation that omits the interaction of the poorly connected bed is presented to establish the explanatory power of the poorly connected bed when coupled to the connected fraction of the distributed system. One interesting outcome of the simulations is the suggestion that anti-phase behaviour between the well- and poorly-connected bed areas may be explained by diurnal variations in cavity opening by sliding, in addition to (or instead of) hydromechanical coupling and the transfer of normal stress, as has been previously assumed in the literature. This idea may have been presented in the observational work, but is corroborated here with the model.

Despite its caveats and some of the less-satisfying ad-hoc aspects of the model (see below), this study is novel and important. Though the scientific conclusion of the work has already been put forth in the

group's observational papers, this is the first attempt (to my knowledge) to include poorly-connected drainage in an otherwise distributed continuum model of ice-sheet hydrology. The message that observations from Greenland demand a more nuanced understanding than the distributed-versus-channelized conceptual model will elevate the conversation on basal drainage and improve our framework for interpreting hydrologically-forced ice dynamics in Greenland.

We thank the reviewer for the careful consideration of this work, and we appreciate the reviewer's acknowledgment of the challenges in developing a model for the weakly-connected system and in creating a suitable comparison between the model and observations. This reviewer raises valid and helpful concerns that we have addressed below.

Specific comments

1. Though the model presented provides an adequate basis for the claims made in the paper, it still has some unsatisfying elements. The authors have, no doubt, painstakingly tried to deal with the lack of direct observational constraints on model formulation and parameters ubiquitous in any attempt to simulate glacier drainage. In contrast to most other theoretical/modelling studies, the authors are able to draw directly upon data specific to the study site. Still, one is left wondering about the extent to which the results are a function of the chosen formulations or parameters, particularly where the authors state that parameters were chosen to achieve a particular result. Among the more ad-hoc aspects of the model, the authors may want to comment on the sensitivity of results to:

(a) A choice of f_w other than $2/3$. This value is justified based on the fact that all of their boreholes tapped into the "weakly connected" drainage system, but how sensitive are the results to, for example, $f_w = 0.5$ or 0.9 ?

This is a good question, and we are able to constrain an approximate upper bound on this parameter from observations and explore the effect of different values using the model. A new section has been added to the Supplementary Information document (Model sensitivity to area fraction of the weakly-connected system) and is referenced in the main text. In this new analysis, we present a back-of-the-envelope calculation for the upper bound of f_w based on observational constraints, and then present results from two new simulations with different f_w values (0.50 and 0.90, which we found were indeed good values to demonstrate the sensitivity of the model to this parameter). Approaching $f_w=0.90$, the results become nonsensical, while approaching 0.50, the qualitative behavior of the model is retained but quantitatively the model performs less well relative to observations of ice speed.

(b) Constant f_w . Could the authors comment on why they chose to keep f_w constant rather than allowing the bed fraction of connected versus unconnected drainage to evolve? I assume the argument against an evolving f_w is that an additional governing equation (of an unknown form) would be required.

A changing the area fraction during summer is a plausible mechanism by which the weakly-connected system could affect ice dynamics. While we cannot rule out the possibility that the area fraction changes during the summer, we found that any substantial change to it causes a significant change in the diurnal range of ice speed, which is not justified by the observations. This, along with the desire to keep the model as simple as possible while retaining the most important observed behavior, is why we did not include these results originally. However, the reviewer's question highlights that the inability of changes in f_w to explain the seasonal decline in ice speed is a valuable result in its own right, and so we have added these results to the manuscript. A description of new model runs and new figures are in the Supplemental Information (Model sensitivity to area fraction of the weakly-connected system), and a short paragraph summarizing the new analysis has been added to the "Modeled ice velocity" of the main text.

(c) The prescribed evolution of the permeability coefficient k_{0w} during the melt season. Does the form of the function matter?

Yes, the form of this function affects the details of the time-series of water pressure in the weakly-connected system. We experimented with a number of functional forms, including having k_{0w} be ad hoc functions of the water thickness or effective pressure within the distributed system. We found that these more complicated parameterizations gave similar results but with differences in the details of the seasonal trajectory of water pressure. Because we found that the key attribute of this governing equation is that the permeability coefficient is greater in summer than in winter, we ultimately chose to use a very simple parameterization. We certainly feel the governing relations for changing permeability is a critical area for further study, should detailed observations become available by which to reasonably assess it. We feel comparing various ad hoc equations for the permeability of the weakly-connected system adds little value to this work because there is so little observational or physical basis by which to construct them. However, we have added text highlighting this gap in knowledge as a critical area for further research (Methods: Weakly-connected drainage model).

Do any of the above (a,b,c) affect the estimated timescales presented at the top of page 6, for example?

The choices of the above have little effect on the estimated recharge timescales because this is estimated based on the basal melt rate (which is independent of the above parameters). The estimated recharge timescale is also controlled by the degree to which the weakly-connected system has drained during summer. While that quantity would be affected by the above parameter choices, we have calibrated it in the model to borehole observations, meaning conclusions on recharge time scale are not sensitive to these choices.

Finally, the simplified model geometry (Figure 2a) contrasts sharply with reality when it comes to bed geometry. It would be reassuring to see some acknowledgment of, or speculation on, the possible influence of real bed topography on the hydrology.

We have added a new paragraph to the “Methods: Model setup” section where the model domain is described that discusses this issue. In short, we acknowledge the importance of bed topography on subglacial hydrology, and justify our simplified geometry because of the uncertainty of the bed topography in our study area. We also point out that because we are primarily interested in model results at a moulin where bed topography is less important in controlling subglacial drainage, our primary results should not be strongly affected by this simplification.

All of these points could be addressed in the Supplementary Material.

2. Success of the coupled simulation versus the control model, Figure 3. This result is central to the claim of the paper: that poorly-connected areas of the bed play a fundamental role in the seasonal dynamics of ice flow. By eye, Figure 2b looks better than Figure 2c in emulating Figure 2a, but not in an overwhelming and obvious way. It would strengthen the authors’ central claim if the superior match of 2b were demonstrated in some quantitative metric. It is really the temporal evolution of the slope of the lines that is important here. Perhaps a correlation coefficient between the modeled and observed versions of such a quantity would more convincingly establish that 2b is a better match than 2c.

Adding a quantitative assessment for model skill is a very good suggestion. We have added text discussing the assessment of both the slope and the intercept of the line segments in Figure 3. See revised text for discussion in sections “Modeled ice velocity”, “Methods: Ice velocity calculations”, and

new figure in Supplementary Information document. We interpret the intercept of the lines in Figure 3 as representing the seasonal evolution of the relationship between moulin hydraulic head and sliding. We show that intercepts of the lines for the model version with an evolving weakly-connected system are significantly positively correlated with those of the observations. The model with the static weakly-connected system has no statistically significant correlation with the observations. We also assess the slopes of the lines in Figure 3. We interpret the slope as being a measure of the sensitivity of sliding to effective pressure as effective pressure changes. This behavior is expected from theory and is clearly exhibited by our observational dataset. We use linear regression to show that this relationship is similar for the observations and both of our model versions, confirming our tuning of parameters in the basal friction law. Notably, this changing relationship explains the varying slopes in Figure 3c (the model with the static weakly-connected system).

3. The authors choose to present the weakly connected bed fraction as a third component of the drainage system, in addition to the “distributed” and “channelized” components. In anticipation of this paper’s influence and uptake, I would encourage the authors to carefully consider their terminology. Channels are “connected” and efficient by nature, as presented in the paper. The distributed system is inefficient by nature (based on the physics), but it is not clear to me that it must also be fully “connected” as presented in the paper. Have the authors considered casting the distributed system as a mixture of poorly- and well-connected fractions (as the model is currently implemented), rather than contrasting the distributed system (assumed inherently connected) with the weakly connected system? The current justification is given on page 7, lines 187-191, but it could equally be the case that previous work has neglected to acknowledge variable connectivity in the distributed drainage system. A related issue is whether the “weakly connected” drainage system includes truly unconnected bed areas. Though just semantics, these labels should be carefully chosen as they are likely to be propagated in through the literature.

We appreciate the reviewer’s concern for accurate terminology in proposing a new component to a longstanding conceptual model. This is actually a topic we worked hard on in the original submission. For example, in earlier versions of this work we had used the previously published term “isolated” instead of “weakly-connected”. However, in feedback from presenting this work in talks at conferences, it became apparent that “isolated” connoted no interaction with the surrounding bed whatsoever, while the weak interactions with the rest of the system are key to the conceptual model we are proposing.

The reviewer’s suggestion that the weakly-connected system may really be a subset of distributed drainage is a valuable clarification to emphasize, and we have added some emphasis on this idea to existing text stating this in the “Conceptual model” section.

From a practical perspective, we believe modeling the weakly-connected areas as a subset of the distributed system with weaker permeability is a feasible alternative approach to the method we have applied here. We originally considered such an approach, but ultimately chose to use a subgrid parameterization of a third component for two reasons. First, because GPS and satellite measurements of ice surface velocity indicate smooth velocity fields, it can be inferred that spatial variability in effective pressure and associated drainage conditions at the bed is at the length scale of an ice thickness or less (i.e., smoothed as “seen” by the ice dynamics). Representing such spatial heterogeneity at the grid scale would require a very fine grid, while the subgrid parameterization allowed us to represent the weakly-connected system at a coarser resolution, meaning our approach may be transferrable to large-scale ice sheet simulations. Second, explicitly modeling variable permeability of the distributed system would require additional assumptions and parameters about the spatial distribution of the system. This discussion has been added to the section “Methods: Weakly-connected drainage model”.

Reviewer #2 (Remarks to the Author):

Review of the article entitled "Greenland subglacial drainage evolution regulated by weakly-connected regions of the bed"

This study investigates the link between the subglacial drainage system of Greenland Ice Sheet and its surface velocities. The novelty of this publication lies in the use of a three component subglacial hydrological model where preceding models were using only two components to describe the subglacial drainage system. The results of the model compared to observations made at a Greenland borehole site show an improvement in the modelling of the subglacial water pressure and hence its influence on the basal velocity of the Ice Sheet.

The Study presented here is clearly written, and sufficiently documented both in terms of data comparison and model presentation. I recommend a publication of this study as is with only a few minor comments listed bellow.

Throughout the manuscript, references to the Methods could be more precise to help find relevant informations (equation numbers, section numbers)

To help guide the reader, we have added specific reference to the section or equation within the Methods in each of the eight references to the Methods in the main text.

Line 15 : "increasing subglacial water pressure" in place of " increasing water pressure"

Change made.

Line 175: I would omit "of the bed"

That text does not occur on line 175. I am interpreting that perhaps line 177 is meant here. The phrase "of the bed" on that line does seem superfluous, so I have removed it.

Line 202: "...time scale of years in our model." rather than "... time scale in our model of years."

Change made.

Caption of Figure 2 : Model domain is presented with thick dashed lines and not thin lines as stated. Cyan is probably not the best colour choice for the equilibrium line.

Caption has been corrected. The cyan line has been changed to purple and the caption updated accordingly.

Line 266 . Missing unit for "p_ice"

Correction made.

Line 341 and 342: Hydraulic heads should be expressed in meters, I guess that the percentage refers to percentage of the flotation head

In an earlier draft of the manuscript water pressures were expressed everywhere as fraction of overburden pressure. In later revisions we changed to using head relative to floatation elevation

because that provided a more accurate and direct comparison to observations, given uncertainties in observed ice thickness and discrepancies in ice thickness between the idealized model domain and the field site. This line was never updated to reflect the new units. The correction has been made.

SI Table 3: There are some exponent that should not be there

Those are actually citations to the papers from which the reference values are taken. To make this clear, I have added the author's name before the citation superscript.

Reviewers' Comments:

Reviewer #1 (Remarks to the Author):

The authors have provided a thorough and thoughtful response to the reviews, and have revised the manuscript accordingly. Congrats on a really nice piece of work.